

# Reactive navigation under a fuzzy rules-based scheme and reinforcement learning for mobile robots

Elizabeth López-Lozada, Elsa Rubio-Espino, J. Humberto Sossa-Azuela and Victor H. Ponce-Ponce

Centro de Investigación en Computación, Instituto Politécnico Nacional, Mexico, Mexico City, Mexico

## ABSTRACT

Robot navigation allows mobile robots to navigate among obstacles without hitting them and reaching the specified goal point. In addition to preventing collisions, it is also essential for mobile robots to sense and maintain an appropriate battery power level at all times to avoid failures and non-fulfillment with their scheduled tasks. Therefore, selecting the proper time to recharge the batteries is crucial to address the navigation algorithm design for the robot's prolonged autonomous operation. In this paper, a machine learning algorithm is used to ensure the extended robot autonomy based on a reinforcement learning method combined with a fuzzy inference system. The proposal enables a mobile robot to learn whether to continue through its path toward the destination or modify its course on the fly, if necessary, to proceed toward the battery charging station, based on its current state. The proposal performs a flexible behavior to choose an action that allows a robot to move from a starting to a destination point, guaranteeing battery charge availability. This paper shows the obtained results using an approach with thirty-six states and its reduction with twenty states. The conducted simulations show that the robot requires fewer training epochs to achieve ten consecutive successes in the fifteen proposed scenarios than traditional reinforcement learning methods exhibit. Moreover, in four scenarios, the robot ends up with a battery level above 80%, that value is higher than the obtained results with two deterministic methods.

# INTRODUCTION

Autonomous mobile robots are getting a great deal of attention due to their adoption in many areas such as space exploration, search and rescue tasks, inspection and maintenance operations, in agricultural, domestic, security, and defense tasks, among many others. They are generally composed of three main modules that allow them to complete their job. The first module integrates all the mechanisms and elements that materialized the robot's locomotion system, including the pneumatic, electromechanical, electrical, and electronic components. The second module deals with the environment's data acquisition based on sensors and the software needed. The third module is the robot's brain, i.e., this module is responsible for data processing, robot control, and navigation.

Corresponding authors
Elizabeth López-Lozada,
elopezl0807@alumno.ipn.mx
Elsa Rubio-Espino,
erubio@cic.ipn.mx

Mobile robot navigation involves any programmed activity that allows the robot to move from its current position to a destination point (*Huskić & Zell, 2019*), being the path planning and obstacle avoidance the main tasks performed during this process. Navigation algorithms commonly use artificial intelligence (AI) to efficiently accomplish their current mission during obstacle avoidance maneuvers. Various AI approaches are employed to deal with the complex decision-making problems faced during mobile robot navigation, such as fuzzy inference systems (FIS), neural networks (NN), genetic algorithms (GA), A⋆ algorithms, and the artificial potential field method (APF) (*Gul, Rahiman & Alhady, 2019*). For these navigation approaches where path planning and obstacle avoidance are sub-tasks commonly solved during navigation time, the employed methodologies assume that the environment corresponds to static scenarios with obstacles and destinations that do not change over time or dynamic scenarios with obstacles and destinations changing over time. Among the different methods reported in the literature, the APF method demonstrates a suitable way of generating paths to guide mobile robots from their initial position to their destination. This method involves using a simple set of equations, easily programmable under limited computing platforms like the mobile robot's embedded processors, rendering an adequate reactive response to the environment.

As far as decision-making is concerned, several variants are employed involving using different methodologies that help in the decision process. When the decision process implies reaching a predefined goal with obstacle avoidance capability, fuzzy logic or bioinspired methodologies can be used for a robot to decide between avoiding an obstacle, following a wall to the right, or to the left (*Zapata-Cortes, Acosta-Amaya & Jimnez-Builes, 2020*; *Khedher., Mziou. & Hadji., 2021*; *Yang, Bevan & Li, 2020*; *Wang, Hu & Ma, 2020*). Within the decision-making process, it is possible to increase precision in parameters such as the speed, or the robot's angle rotation, improving the executed movements in narrow paths (*Teja S & Alami, 2020*). However, these navigation methodologies fail to consider battery recharging, which is vital to keep a robot working for prolonged periods. Since robots use batteries to power their motors and the navigation system, the robot's battery must supply adequate voltage and current levels for the robot to complete its tasks without any unexpected energy interruption due to a low battery condition. This problem is known as the autonomous recharging problem.

In recent years, the number of applications using mobile robots is increasing, and linked with this growth is the autonomous operation margin. The autonomous recharging problem (ARP) is getting attention to effectively planning and coordinating when, where, and how to recharge robots to maximize operational efficiency, improving the robot's autonomy. ARP means all the actions required to decide the moment, the destination, and how to recharge the robot's battery to maximize operational efficiency (*Tomy et al., 2020*). The ARP faces two main problems: how to proceed to the battery charge station and the precise moment to deviate from the main path to the final destination. The first deals with designing and testing the hardware and software employed to help the robot maneuver to the charging station. The second involves deciding when the robot should go to the charging station (*de Lucca Siqueira, Della Mea Plentz & De Pieri, 2016*). This work focuses only on the second problem.

This proposal introduces a novel approach to solve the ARP based on machine learning to determine the robot's pertinent moment to recharge its battery. The proposed system consists of a path planning module and a decision-making module. The decision-making module uses the fuzzy Q-learning (FQL) method for the robot to decide between going to the destination point or head to the battery charging station or remain static. The paper's main contribution lies in a reactive navigation scheme that grants a robot to learn, based on trial and error, to make decisions to fulfill its tasks with a suitable battery level. The following sections describe the related work, as well as the analyzed methods and the proposed approach.

## Related work

One of the classical methods used to address the path planning problem is the APF method, which bases its foundations on attractive and repulsive forces. The APF method has been widely applied in static real-time path planning. Several works found in literature addressing the APF method make slight modifications to Khatib's model, introduced in 1985, to improve its performance and avoid falling into a minimum local state. For example, the work of *Hosseini Rostami et al. (2019)*, which uses APF method in a dynamic environment, showed robot obstacle avoidance capability while moving towards its target (*Matoui et al., 2017*). It implemented the APF method and combined a set of equations to operate mobile robots cooperatively under a decentralized architecture. Alternatively, this method's implementation combined with fuzzy logic (*Tuazon et al., 2016*) and reinforcement learning (RL) (*Liu, Qi & Lu, 2017*) is another approach employed for robot navigation. Other alternatives to solve the path planning are addressed using NN (*Wei, Tsai & Tai, 2019*) or FQL (*Lachekhab, Tadjine & Kesraoui, 2019*).

Some strategies have arisen to solve the ARP focus on when and where to recharge the battery. The traditional charging method refers to move automatically to a charging station when the battery charge is below a certain threshold. It is a simple strategy to solve the ARP, but with little flexibility, i.e., the robot could be about to reach the goal, but instead, it heads to charge the battery by considering only a charge level threshold as the only criteria. To solve this disadvantage (*Rappaport & Bettstetter, 2017*) proposed using an adaptative threshold using five policies to select a charging point (CP). The first policy consists on select the closest CP, the second is to select a free CP, the third policy is to choose a CP when there are not more possibilities to explore, the fourth consists of stay as long as possible before moving to the next one, and the last policy refers to send information to other robots to avoid redundancies in exploration.

Based on a fuzzy inference system (*de Lucca Siqueira, Della Mea Plentz & De Pieri, 2016*) propose a solution for the first sub-problem. It considered three crisp input variables, the battery level, the distance to the charging station, and the distance to the destination point for the fuzzy mapping rules. Under this approach, if a warning state is set-on due to a sensed low battery level, and the destination point is closer than the charging station, the robot will head towards the target, and it will not turn off. An alternative way of resolving ARP is the scheduling strategy of mobile chargers (MC). The MCs improve the battery charging process while the robots perform a task, avoiding task execution failure due to a low battery

level and prioritizing the task completion. It is necessary to establish a minimum number of MCs and a charging sequence algorithm to design an operative scheduling strategy. Some methods used in designing a scheduling strategy are linear programming, nonlinear programming, clustering analysis, TSP algorithm, queuing theory, and other mathematical methods. In this way, *Cheng et al. (2021)* proposed a mesh model to cope with the robot's limited movements and the MCs. Considering the distances between the MCs and the robot and the expended time at the charge station, they developed a scheduling algorithm of minimum encounter time to solve the recharging problem in a robot that executes a priority task. *Tomy et al. (2020)* proposed a recharging strategy based on a learning process to schedule visits to the charging station. The main contribution reported was the robot's capability to predict high-value tasks assigned for execution. Therefore, the robot can find the specific moment to schedule a visit to a charging station when it is less busy and does not have essential tasks to execute. This proposal could have more flexibility compared to a system based on rules only.

Another solution to the ARP is task planning. The main idea is that after solving a task, the robot goes to a charging station and recharges its battery. In this way, *Rappaport & Bettstetter (2017)* proposed a method where a robot has a sequence of tasks, and it has to segment the sequence to plan when to recharge. In the second stage, multiple robots were coordinated to recharge using a maximum bipartite graph matching method. Similarly, *Xu, Wang & Chen (2019)* describe a method to solve the ARP with multiple robots capable of planning regular visitings to the charging station as part of a sequence of tasks they must follow under a collaborative environment.

## Methods

This section summarizes the methods used in this work for the navigation approach, starting with the classical reinforced learning methods: Q-learning (QL) and the State-Action-Reward-State-Action (SARSA) algorithm. This section ends up with the fuzzy Q-Learning (FQL) method description implemented in the decision-making module.

### The artificial potential field method

The APF method provides a simple and effective motion planning method for a practical purpose. Under this method, the attractive force is computed according to Eq. (1):

$$F_{attr}(q,g) = -\xi p(q,g) \tag{1}$$

where $p(q,g)$ is the euclidean distance between the robot's position, denoted by $q$, and the destination point's position $g$. The term $\xi$ is defined as the attractive factor. The repulsive force, on the other hand, is calculated with the aid of Eq. (2).

$$F_{rep}(q) = \begin{cases} \eta(\dfrac{1}{p(q,q_{o_j})} - \dfrac{1}{p_{o_j}})\dfrac{p^2(q,q_{o_j})}{p(q,q_{o_j})}, & \text{if } p(q,q_{o_j}) < p_{o_j} \\ \eta(-\dfrac{1}{p_{o_j}}), & \text{if } p(q,q_{o_j}) = p_{o_j} \\ 0, & \text{if other case} \end{cases} \tag{2}$$

where $p(q,q_{o_j})$ is the distance between the robot's position and the $j$th obstacle's position $q_{o_j}$. The obstacle radius threshold is denoted by $p_{o_j}$. The term $\eta$ is the repulsive factor.

Furthermore, the resultant force Eq. (3) is the sum of attractive and repulsive forces.

$$F_{res} = F_{attr}(q, g) + F_{rep}(q) \tag{3}$$

### Classical reinforcement learning methods

Reinforcement learning (RL) is a machine learning paradigm, where the main identified elements are the agents, the states, the actions, the rewards, and punishments. In general, RL problems involve learning what to do and how to map situations to actions to maximize a numerical reward signal when the learning agent does not know what steps to take. The agent involved in the learning process must discover which actions give the best reward based on a trial and error process (*Sutton & Barto, 2018*). Essentially, RL involves closed-loop problems because the learning system's actions influence its later inputs. Different methods use this paradigm; among the classic methods are QL and SARSA. QL provides learning agents with the ability to act optimally in a Markovian domain experiencing the consequences of their actions, i. e., the agents learn through rewards and punishments (*Sutton & Barto, 2018*). QL is a method based on estimating the value of the $Q$ function of the states and actions, and it can be defined as shown in Eq. (4), where $Q(S_t, A_t)$ is the expected sum of the discounted reward for the performance of action $a$ in a state $s$, $\alpha$ is the learning rate, $\gamma$ is the discounting factor, $S_t$ is the current state, $S_{t+1}$ is the new state, $R_{t+1}$ is the reward in the new state, and $A_t$ is the action in $S_t$, and $a$ is the action with the best $q$-value in $S_t$.

$$Q(S_t, A_t) \leftarrow Q(S_t, A_t) + \alpha[R_{t+1} + \gamma \max_a Q(S_{t+1}, a) - Q(S_t, A_t)] \tag{4}$$

The method called SARSA results from a modification made to QL. The main difference between these algorithms is that SARSA does not use the max operator from the Q function during the rule update, as shown in Eq. (5) (*Sutton & Barto, 2018*).

$$Q(S_t, A_t) \leftarrow Q(S_t, A_t) + \alpha[R_{t+1} + \gamma Q(S_{t+1}, A_{t+1}) - Q(S_t, A_t)] \tag{5}$$

### Fuzzy Q-learning method

FQL (*Anam et al., 2009*) is an extension of fuzzy inference systems (FIS) (*Ross, 2010*), where the fuzzy rules define the learning agent's states. At the first step of the process, crisp input variables are converted into fuzzy inputs through input membership functions. Next, rule evaluation is conducted, where any convenient fuzzy $t$-norm can be used at this stage, a common choice is the *MIN* operator. Only one fuzzy rule, i.e., the $i$th fuzzy rule, $r_i$, determines the learning agent's state, and $i \in [1, N]$, $N$ denotes the total number of fuzzy rules. Each rule has associated a numerical value, $\alpha_i$, called the rule's strength, defining the degree to which the agent is in a certain state. This value $\alpha_i$ allows the agent to choose an action from the set of all possible actions $A$, the $j$th possible action in the $i$th rule is called $a(i, j)$, and its corresponding $q$-value is $q(i, j)$. Therefore, fuzzy rules can be formulated as:

**If** $x$ is $S_i$ **then** $a(i, 1)$ with $q(i, 1)$ **or** ... **or** $a(i, j)$ with $q(i, j)$

FQL's primary goal implies the learning agent finds the best solution for each rule, i.e., the action with the higher $q$-value. This value is rendered from a table of $i \times j$ dimensions, containing $q$ number of values. The table dimension corresponds to the number of fuzzy rules times the total number of actions. A learning policy selects the right action based on the quality of a state-action pair, based on the Eq. (6), where $V(x, a)$ is the quality value of the states, $i^*$ corresponds to the optimal action index, i.e., the action index with the highest $q$-value, and $x$ is the current state. On the other hand, the exploration-exploitation probability is given by $\varepsilon = \frac{10}{10+T}$, where $T$ corresponds to the step number, is assumed in this work.

$$V(x, a) = \frac{\sum_{i=1}^{N} \alpha_i(x) \times q(i, i^*)}{\sum_{i=1}^{N} \alpha_i(x)} \tag{6}$$

The functions in Eq. (7) allow to get the inferred action and its corresponding $q$-value, where $x$ is the input value in state $i$, $a$ is the inferred action, $i^o$ is the inferred action index, $\alpha_i$ is the strength of the rule and $N$ is a positive number, $N \in \mathbb{N}^+$, which corresponds to the total number of rules.

$$a(x) = \frac{\sum_{i=1}^{N} \alpha_i \times a(i, i^o)}{\sum_{i=1}^{N} \alpha_i(x)}; \quad Q(x, a) = \frac{\sum_{i=1}^{N} \alpha_i \times q(i, i^o)}{\sum_{i=1}^{N} \alpha_i(x)} \tag{7}$$

Also, it is necessary to calculate an eligibility value, $e(i, j)$, during the $q$-value updating phase by means of Eq. (8),

$$e(i, j) = \begin{cases} \lambda \gamma e(i, j) + \dfrac{\alpha_i(x)}{\sum_{i=1}^{N} \alpha_i(x)} & \text{if } j = i^o \\ \lambda \gamma e(i, j) & \text{other case} \end{cases} \tag{8}$$

where $j$ is the selected action, $\gamma$ is the discount factor $0 \leq \gamma \leq 1$ and $\lambda$ is the decay parameter in the range of $[0, 1]$. Next, the Eq. (9) is used for the $\Delta Q$ calculation, where $r$ corresponds to the reward.

$$\Delta Q = r + \gamma V(x, a) - Q(x, a). \tag{9}$$

And finally, Eq. (10) updates $q$-value, where $\epsilon$ is a small number, $\epsilon \in (0, 1)$, which affect the learning rate,

$$\Delta q(i, j) = \epsilon \times \Delta Q \times e(i, j). \tag{10}$$

## Proposed hardware

The documented results come from navigation simulations, only. The testing results on a physical robot are left for future work. However, this section presents the hardware proposal developed to carry out the navigation proposal and explains some technical details to be considered during the implementation. The proposed hardware shown in Fig. 1 is the King Spider robot from the BIOLOID Premium kit of ROBOTIS (*ROBOTIS, 2019*). This robot is configured with six legs with a total of 18 servo-motors with 3 degrees of freedom per leg. Each actuator uses the AX-12A servo-motors, which have mobility from

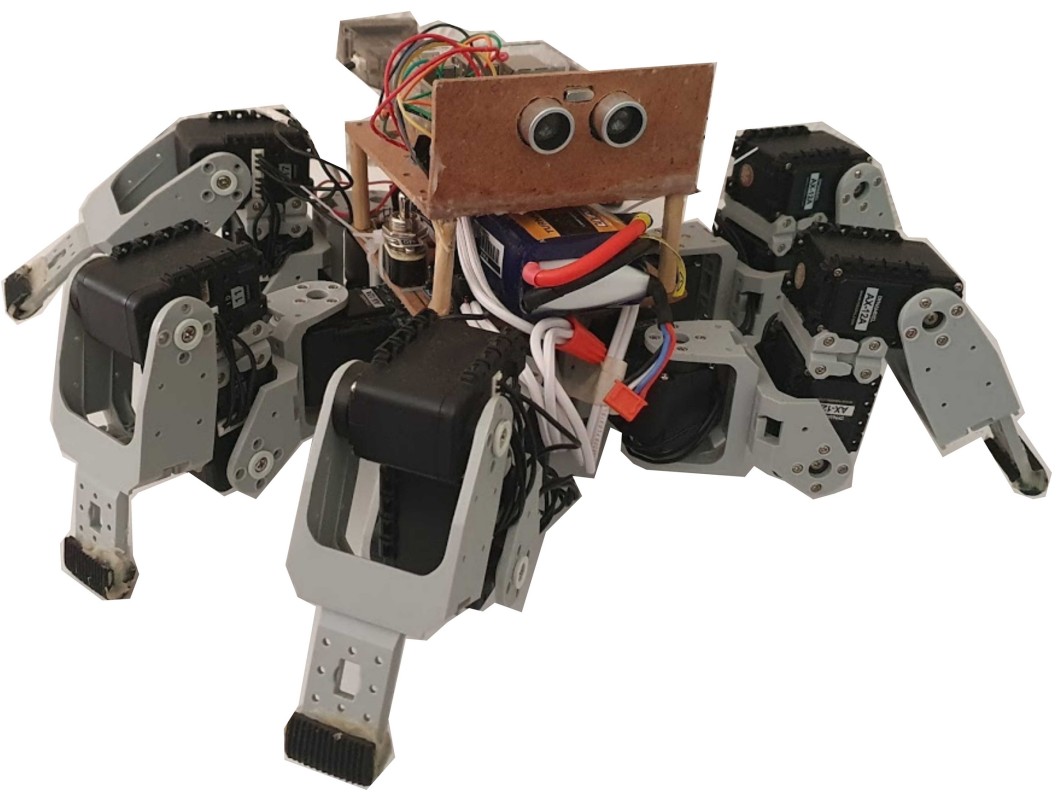

**Figure 1** The King Spider robot, equipped with: 3DOF per leg raspberry pi 3, GPIOS, INA260 and HC-SR04 ultrasonic sensor.

0° to 360°, equipped with a serial interface to establish communication with the processing unit, a microcontroller Raspberry Pi, running the Raspbian Buster operating system. The microcontroller has an I²C interface for energy monitoring and a pair of GPIOS to connect the ultrasonic sensor. The INA260 high-accuracy current and power monitor were selected to measure the battery charge level and the power consumption. For detecting obstacles, the HC-SR04 ultrasonic sensor was chosen, which allows detecting objects at a distance of 40 cm.

The power source considered for use with this robot is an 11.1v Li-Po 3S battery with a capacity of 4250 mAh. Figure 2 shows in solid line the voltage level versus charge percentage, named as battery discharge curve, employed to estimate the remaining robot battery charge (*Tom, 2019*). For simulation, the expression $BL(t) = -1.8245t + 100$ is implemented, where $BL$ corresponds to the battery voltage level in volts, and $t$ is the corresponding time step. The dashed and doted line in Fig. 2 shows the battery discharge approximation given by $BL(t)$.

This hardware was selected to employ it in exploration tasks or as a service robot, taking advantage of its extremity's ability to travel in irregular terrains. However, other robotic platforms, as differential robots, may also be used. The selection of the robotic platform is left to the context specifications. This work's specifications involve designing software

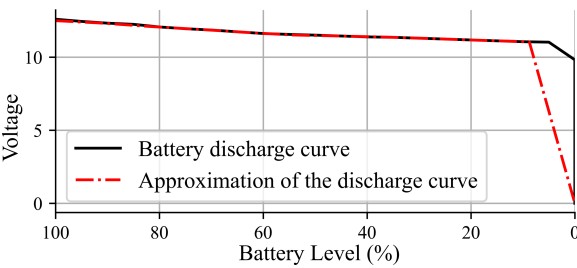

**Figure 2  Battery discharge curve.**

to run on a Raspberry Pi with a Linux-based operating system. The proposal seeks to be modular. Therefore, a path planning module and a decision-making module are handled. The movements were limited to four movements: forward, backward, left, and right within the path planning module. The generated route can be saved in a text file. It is suggested to add a control module that contains the robot's kinematics, receiving as inputs the next position where the robot has to move. In case the control module requires other parameters such as speed or direction angles, or if kinematics has to be adapted to a determined robot configuration, the software provided in this paper would have to be modified. The simplest way would be to call a function that interprets the coordinates within the control module to obtain the required parameters. One of the variables that could be affected if robotic configuration changes is the battery charge estimation, as other robotic platforms could occupy batteries with different features. Therefore, the system's behavior could change according to the discharge range. This approach uses separated modules from the program's main body. The input values come from functions that update and normalize the battery level and the distances to avoid significant modifications to the proposal in the implementation with other hardware.

## BACKGROUND

The proposed navigation system comprises three main modules, one for the control of the robot, a module for path planning, and another module for decision making. This section begins with a description of the path planning module, using the APF method for this proposal. Then it continues with a description of the decision-making module.

### Path planning module

This module implements the APF method, and it considers an operation under environmentally controlled conditions. This work employs a static environment with one battery charging station, obstacles, and a defined destination point. The workspace is a $10 \times 10$ grid, and each space of the grid is equivalent to one step of the robot. The obstacles were arbitrarily distributed into the workspace to define diverse scenarios. For this module, the attraction factor used in Eq. (1) has a value equal to 2.3, while the repulsion factor in Eq. (2) is 61.5. The robot's movements were limited to left, right, forward, and backward. An objects' position is equivalent to a coordinate pair $(x, y)$. Whereas the robot's position

is equivalent to the coordinate pair $(x_{robot}, y_{robot})$, and when the robot experience a position change, its coordinates increase or decrease one unit. Algorithm 1 shows the steps followed by this module for the path generation.

---

**Algorithm 1:** Path planning

---

robot_pos ← insert the current robot position;
destination_pos ← insert destiny or battery charging station;
obstacles_pos ← insert obstacles positions;
position;
path_pos_list ← initialize an empty list;
res_force_list ← compute the resultant force in every space of the grid;
i ← 0;
**while** *robot_pos != dest_pos* **do**
    path_pos_list[i] ← from the neighborhood obtain the position corresponding to
      the value of the highest resultant force;
    robot_pos ← update with path_pos_list[i];
    i++;
return path_pos_list;

---

Algorithm 1 handles the term neighborhood, which refers to the coordinate pairs $(x_{robot}, y_{robot} + 1)$, $(x_{robot}, y_{robot} - 1)$, $(x_{robot} + 1, y_{robot})$, and $(x_{robot} - 1, y_{robot})$. When the path generation concludes, the result is a list of the robot's coordinates to reach the destination. This module requires the robot's information, i.e., the robot's position, destination, and obstacles positions. The module plans a path to the destination with the known information. Diverse scenarios used during path planning are depicted in Fig. 3, which shows fifteen different stages with scattered obstacles, where the squares represent the obstacles, the star is the destination, and the crossbar is the charging station. There are two paths planned for each of these scenarios: the path to the destination and the path to the charging station, identified with two lines, one linked with circles and the other one linked with triangles.

## Decision-making module

With the proposed decision-making module, we demonstrate that the mobile robot can appropriately choose between heading to the goal, going to the battery charging station, or remaining static. This module comprises three main sections: fuzzification of the input variables, fuzzy rule evaluation, and the selection of the action to execute. Figure 4 depicts an overall view of the proposed system's architecture. Given a robot is powered by a battery, and it travels from a starting to a destination point, it is a fact that the battery loses electric charge during displacement. Then, the battery voltage level (BL) is an input to the fuzzy inference system. Fuzzy rules can be formulated, using common sense, as follows. If the battery level is low and a charging station is nearby, the robot could select to go to the charging station to recharge the battery. Even so, if the robot is far from the charging station but close to the destination point, reaching the destination point would prioritize instead of changing its course to the charging station. Ergo, the distance between the robot

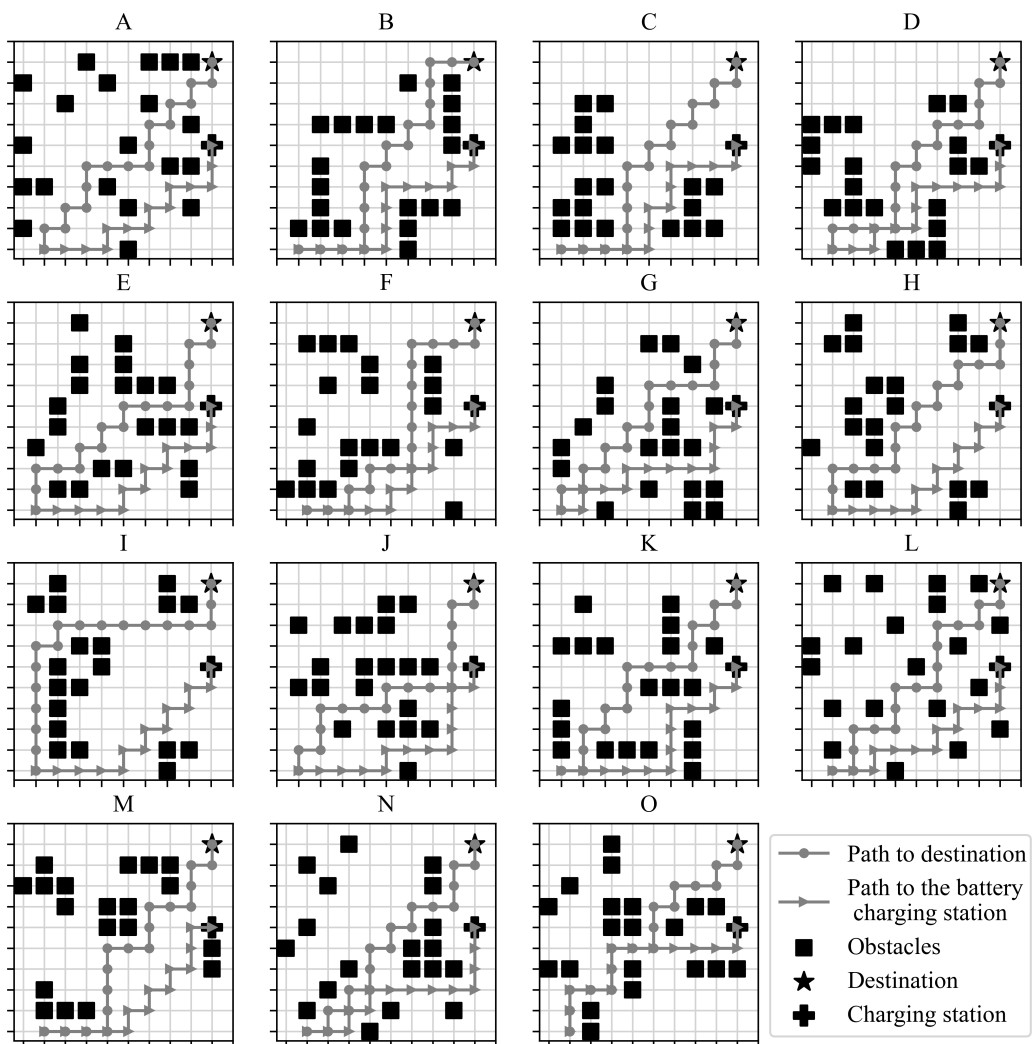

**Figure 3** **Path planning in diverse scenarios.** (A-O) The scenarios used for the simulations showing the routes generated, in each scenario, from the starting point to the battery charging station and destination.

and the destiny is the second input, and the distance between the robot and the charging station is considered the third input.

$$DRD = min\left\{100 \times \frac{CDD}{IDD}, 100\right\}; \quad DRC = min\left\{100 \times \frac{CDC}{IDC}, 100\right\} \quad (11)$$

By using *DRD* in Eq. (11), the distance values are normalized to the value interval of [0, 100], where *DRD* is the distance between the robot and destination, *CDD* is the current distance to the goal, and *IDD* is the initial distance to the goal. Similarly, the system normalizes the distance between the robot, and the charging station with the equation *DRC* in Eq. (11), where *DRC* is the distance between the robot and charging station, *CDC* is the current distance to the charging station, and *IDC* is the initial distance to the charging

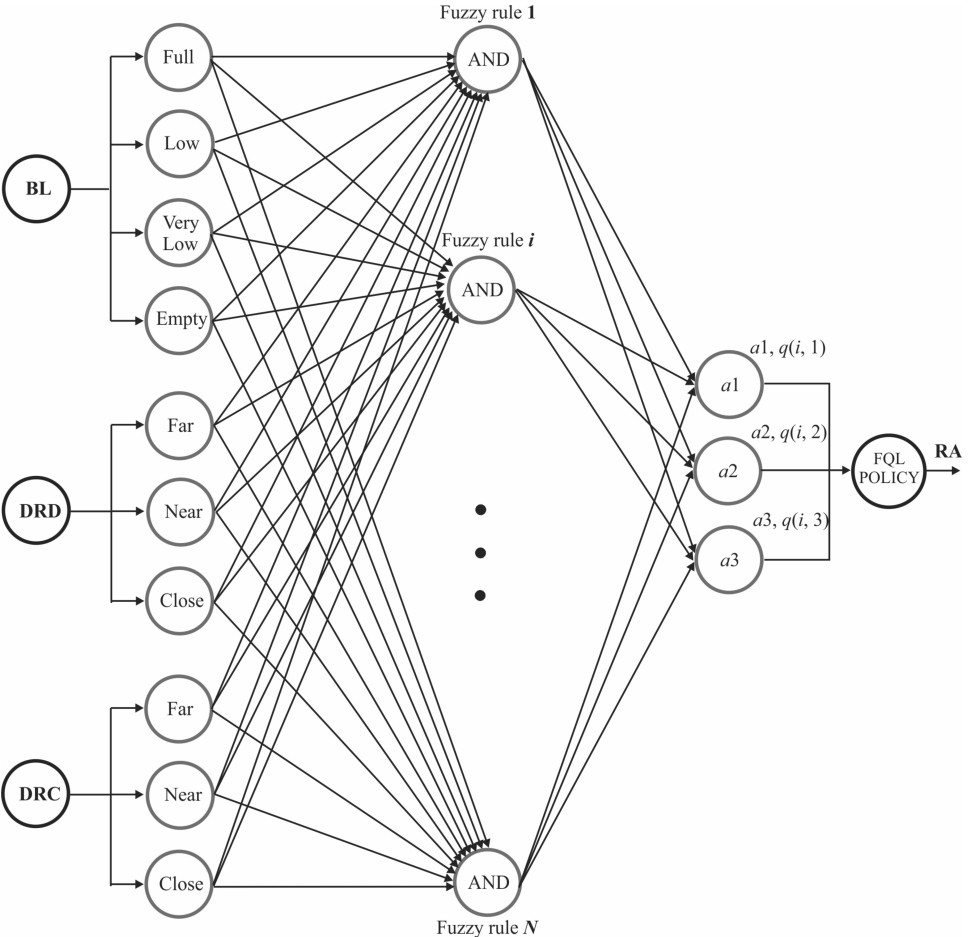

**Figure 4  FQL architecture for decision making.**

station. Simultaneously, the battery level value corresponds to a percentage value between 0 and 100%.

The first input variable, BL, has four fuzzy sets, labeled as Empty, Very Low, Low, and Full. The second input variable, DRD, has three fuzzy sets defined as Close, Near, and Far. Finally, the third input variable, DRC, has three fuzzy sets: Close, Near, and Far. Figure 5 depicts the three input variables with their corresponding triangle-type fuzzy sets. By considering all the possible combinations in the rule's antecedents, the fuzzy system could operate with a total of 36 fuzzy rules, herein named FQL-36. Each rule is formulated to perform a specific robot action (RA) in response to the observed inputs.

Actions denoted by $a1$, $a2$, and $a3$ are associated with a numerical value $q(i, j)$, where the index $i$, corresponds to the $i$th fuzzy rule, and $j$ is the index of the $j$th action. Action index 1 refers to going to the destination point, action index 2 refers to going to the charging station, and action index 3 means to remain static. The action selection is made with Algorithm 2. Likewise, the fuzzy actions are implemented by three output singleton-type fuzzy sets, as depicted in Fig. 6. The exploring-exploitation policy described in the FQL

---

**Algorithm 2:** Action selection

r_pos ← insert *R* position;

d_pos ← insert *D* position;

bcs_pos ← insert *BCS* position;

obs_pos_list ← initialize an empty list;

state ← get current rule;

action ← select an action;

output ← compute with function a(x) in eq. (7);

q ← compute the q value with function q(x) in eq. (7); new_state ← get the current rule;

reward ← get the reward from (12);

states_value ← compute with the eq. (6));

$\Delta Q$ ← compute with eq. (10);

eligibility ← get value from eq. (8);

new_q ← compute the new q with eq. (10));

Update the Q value in the q-table;

return the action;

---

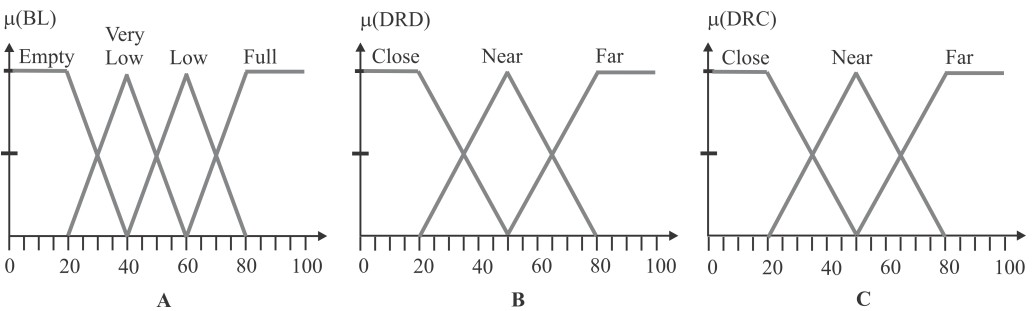

**Figure 5   Input variables and fuzzy sets.** (A) The fuzzy sets of the BL input, where the x-axis is the battery level from 0 to 100, while the y-axis is the degree of membership. (B) The fuzzy sets of the DRD input, where the x-axis is the distance from 0 to 100, while the y-axis is the degree of membership. (C) The fuzzy sets of the DRC input, where the x-axis is the distance from 0 to 100, while the y-axis is the degree of membership.

section serves to make the decision and choose an action. Expression Eq. (12) gives the reward function used, the discount factor, and the learning rate is equal to 0.5 and 0.01, respectively.

$$r = \begin{cases} +10 & \text{if robot is close to destination} \\ +5 & \text{if robot is close to the charging station} \\ -20 & \text{if robot is far of destinations and battery level is very low} \\ -1 & \text{other case} \end{cases} \qquad (12)$$

Given that some rules are redundant and that the fuzzy inference system may not fire any of them, some fuzzy rules are not considered, reducing the fuzzy rule database from

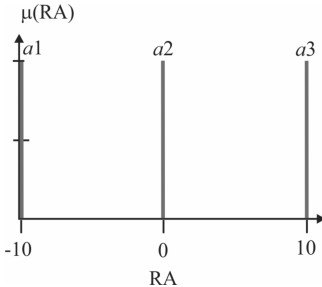

**Figure 6** **Robot's action (RA), output variable with singleton-type fuzzy sets defined.** Fuzzy singleton a1 is the action for heading to the goal, a2 refers to going to the charging station, and action a3 means to remain static.

**Table 1** **The twenty rules of the reduced FIS used in the FQL-20.**

| Rule | BL | DRD | DRC | Output |
|---|---|---|---|---|
| 1 | Full | AC | AC | action =a1 \|a2 \|a3 with q_value =q(1,1) \|q(1,2) \|q(1,3) |
| 2 | Low | Far | Far | action =a1 \|a2 \|a3 with q_value =q(2,1) \|q(2,2) \|q(2,3) |
| 3 | Low | Far | Near | action =a1 \|a2 \|a3 with q_value =q(3,1) \|q(3,2) \|q(3,3) |
| 4 | Low | Far | Close | action =a1 \|a2 \|a3 with q_value =q(4,1) \|q(4,2) \|q(4,3) |
| 5 | Low | Near | Far | action =a1 \|a2 \|a3 with q_value =q(5,1) \|q(5,2) \|q(5,3) |
| 6 | Low | Near | Near | action =a1 \|a2 \|a3 with q_value =q(6,1) \|q(6,2) \|q(6,3) |
| 7 | Low | Near | Close | action =a1 \|a2 \|a3 with q_value =q(7,1) \|q(7,2) \|q(7,3) |
| 8 | Low | Close | Far | action =a1 \|a2 \|a3 with q_value =q(8,1) \|q(8,2) \|q(8,3) |
| 9 | Low | Close | Near | action =a1 \|a2 \|a3 with q_value =q(9,1) \|q(9,2) \|q(9,3) |
| 10 | Low | Close | Close | action =a1 \|a2 \|a3 with q_value =q(10,1) \|q(10,2) \|q(10,3) |
| 11 | Very Low | Far | Far | action =a1 \|a2 \|a3 with q_value =q(11,1) \|q(11,2) \|q(11,3) |
| 12 | Very Low | Far | Near | action =a1 \|a2 \|a3 with q_value =q(12,1) \|q(12,2) \|q(12,3) |
| 13 | Very Low | Far | Close | action =a1 \|a2 \|a3 with q_value =q(13,1) \|q(13,2) \|q(13,3) |
| 14 | Very Low | Near | Far | action =a1 \|a2 \|a3 with q_value =q(14,1) \|q(14,2) \|q(14,3) |
| 15 | Very Low | Near | Near | action =a1 \|a2 \|a3 with q_value =q(15,1) \|q(15,2) \|q(15,3) |
| 16 | Very Low | Near | Close | action =a1 \|a2 \|a3 with q_value =q(16,1) \|q(16,2) \|q(16,3) |
| 17 | Very Low | Close | Far | action =a1 \|a2 \|a3 with q_value =q(17,1) \|q(17,2) \|q(17,3) |
| 18 | Very Low | Close | Near | action =a1 \|a2 \|a3 with q_value =q(18,1) \|q(18,2) \|q(18,3) |
| 19 | Very Low | Close | Close | action =a1 \|a2 \|a3 with q_value =q(19,1) \|q(19,2) \|q(19,3) |
| 20 | Empty | AC | AC | action =a1 \|a2 \|a3 with q_value =q(20,1) \|q(20,2) \|q(20,3) |

36 to 20; herein, named FQL-20. Then the reduced number of fuzzy rules are defined as shown in Table 1, where AC corresponds to any case, and the symbol | refers to logic function OR.

## Navigation system proposal

The modules described so far integrate the robot's navigation system. The system plans two paths to conduct the robot to the goal and the other one to proceed to the battery charging station. If the robot encounters an obstacle during its movement, it re-plans its path to the current destiny. If the robot's action changes between going to the goal or the

charging station or vice versa, it plans a new course if necessary. These changes depend on the results of the decision-making module. The following steps in Algorithm 3 describe the navigation proposal.

---

**Algorithm 3:** Navigation

---

r_pos ← initial position;
d_pos ← insert *D* position;
bcs_pos ← insert *BCS* position;
Initialize obs_pos_list;
destiny ← d_pos;
Generate the MF's and rules;
D_path, BCS_path← generate the paths;
**while** *r_pos != destiny* **do**
  action ← get from action selection;
  next_destiny ← update the destiny;
  **if** *next_destiny != destiny* **then**
    Update the path to next_destiny;
    destiny ← next_destiny
  Update next_r_pos ;
  **if** *there is an obstacle in next_r_pos* **then**
    Set obstacle in obs_pos_list ;
    Update the D_path, and BCS_path;
  **else**
    Move to new position;
    Update r_pos;

---

## RESULTS

This section introduces the simulated results obtained from this navigation proposal. The system simulation with 36 fuzzy-rules is denoted as FQL-36, and the reduced proposal with only 20 rules is referred as FQL-20. The proposal implementation was programmed using Python 3.8 with the PyCharm Community Edition IDE on a computer with an Ubuntu 20.04 operating system. The source code designed for this research can be obtained at https://github.com/ElizBth/Reactive-Navigation-Under-a-Fuzzy-Rules-Based-Scheme-and-Reinforcement-Learning. This proposal can be implemented from scratch by employing the equations related to the APF and fuzzy *q*-learning section, following Algorithms 1, 2 and 3 of the Background Section. Some methods may serve to compare this proposal's performance. Firstly, with a simple two threshold-based method, where the input is the battery level: the first threshold is 40%, so, if the battery level is above this value, the selected action is a1; the second threshold is 20%, so, if the battery level is above this value the selected action is a2; otherwise the selected action is a3. Secondly,

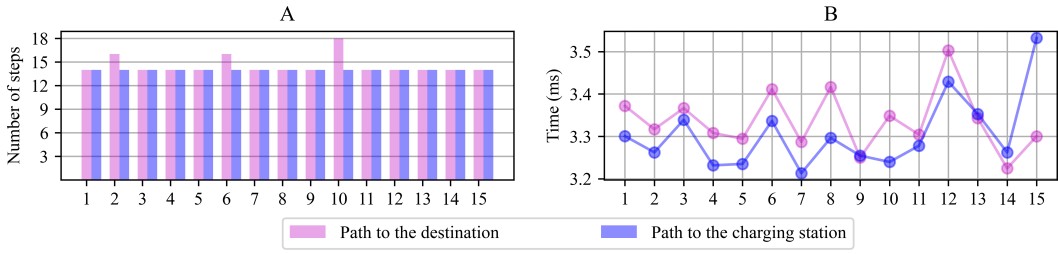

**Figure 7** Number of positions computed to reach the destination or the charging station from the point of departure (A) and computing time lag observed for path planning (B).

being the FQL method resulting from the combination of the QL method with a FIS, they are a good starting point to compare these methods' performance with the introduced proposal. Like the FQL proposal, this work uses two FIS, denoted as FIS-36 and FIS-20. Moreover, the last method considered is the SARSA method, which is similar to the QL method. The inputs of these methods correspond to the battery level and the distances to the destination and the charging station; additionally, the RL methods occupy the reward function expressed in Eq. (12).

## Path planning simulations

Figure 7A shows the planned number of steps to reach the goal for the 15 scenarios proposed in Fig. 3. The number of movements the robot must complete reaching the destination or the charging station, starting from the departing point, is similar in all cases, resulting in average planned paths of 14 positions. Figure 7B shows the module's computing time lag since it performs the parameter initialization until it finds a path free from collisions. The average period consumed to generate the path in each scenario was registered between 3.2 and 3.8 ms from this graph. The navigation system's first planned paths serve during the early navigation stages under each scenario since the system is prone to change direction during the path. The new computations take as reference the new current position to calculate a new path.

## System simulations

The variables used to model the system behavior are the accumulated reward and the number of steps taken to complete the robot's task in each scenario. The simulations initiate with the battery at the maximum charge level. Figure 8 shows the accumulated reward obtained at each scenario after reaching ten task successes. By comparing the performance between FQL-36 and FQL-20, the reward does not significantly change in either case. There are some slight behavior variations in the choice actions, which may have been random or greedy.

There is similar behavior only when the tenth success was reached by observing the accumulative reward. Figure 9 shows the accumulated reward in this attempt with QL, SARSA, FQL-36, and FQL-20. However, there is a more significant difference in behavior when comparing with QL and SARSA; this indicates that there were more cases where the simulation was close to the destination or the charging station, and if there were more

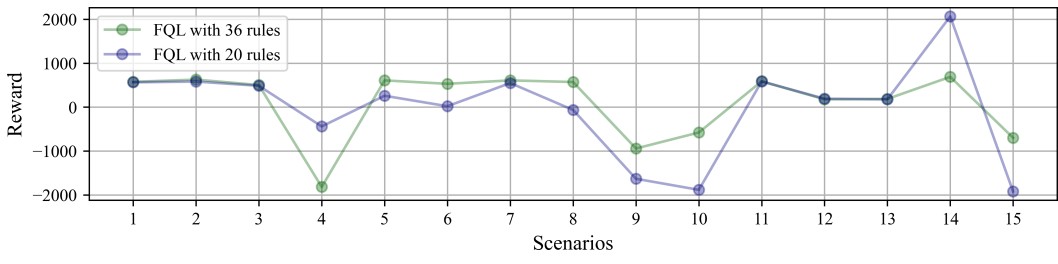

**Figure 8** Accumulated reward obtained at each scenario under simulated test after reaching ten task successes.

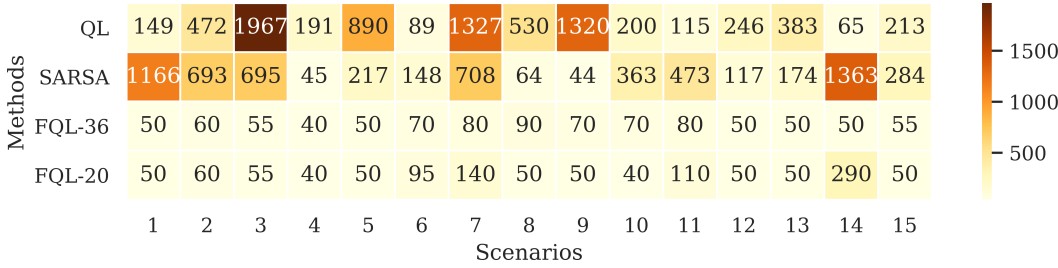

**Figure 9** Accumulated reward on tenth success.

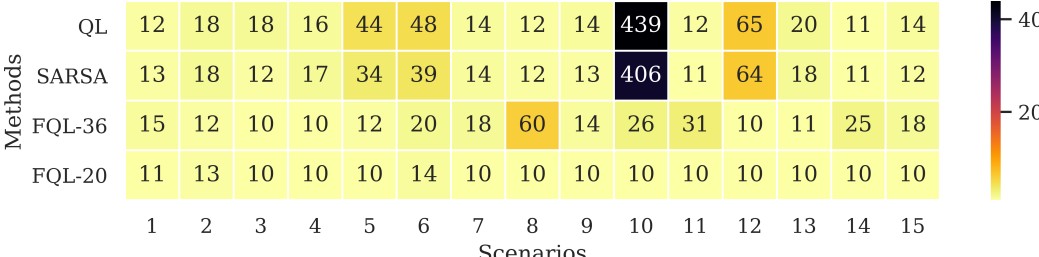

**Figure 10** Number of epochs needed to reach the ten successes.

steps executed, it is logical to see a more generous amount of reward than with FQL-36 and FQL-20. To confirm this behavior, it should observe the number of steps executed during the simulation and the number of epochs it took to reach the destination.

Figure 10 shows the number of epochs it took for the QL, SARSA, FQL-36, and FQL-20 methods. Here, FQL-20 presents better results than the other methods since. In 80% of the scenarios, it only requires ten epochs to reach the ten successes followed by FQL-36, while the other two methods take more epochs to reach the goal. Given these results, this paper's proposal already presents an advantage over classical RL methods by requiring fewer epochs for the system to learn which decisions to make to complete its goal. Figure

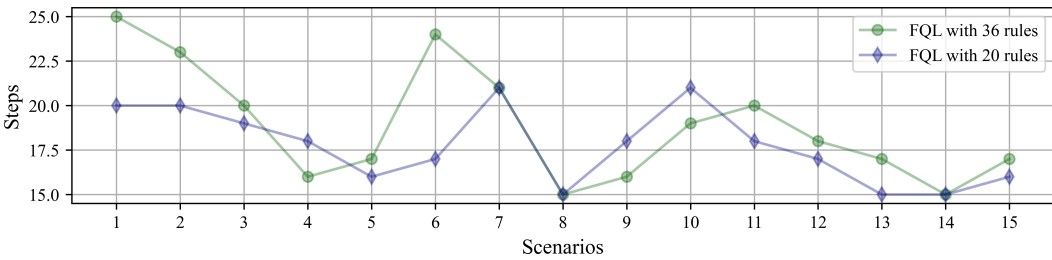

**Figure 11  Number of simulated steps involved in reaching the goal by using any of the systems driven by 32 or 20 fuzzy rules.**

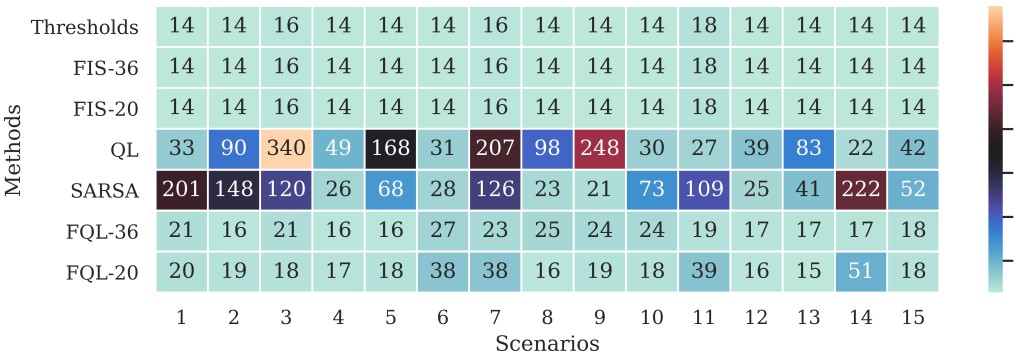

**Figure 12  Number of steps executed on the 10th success.**

10 does not include the threshold-based method or the FIS as they do not require a learning stage.

Figure 11 shows the average number of steps involved in reaching the goal, whose average resulted from the ten successes obtained by using any of the systems driven by 36 or 20 fuzzy rules. The results show that only a difference of three steps was obtained in two scenarios. It is also interesting to verify that the system's overall behavior is better with the base of 20 fuzzy rules. Therefore, the simplicity in the design of the FIS agrees with the effectiveness of the results. Figure 12 shows the comparison with the number of executed steps to obtain the tenth success with the methods mentioned above. Similarly to the epochs, the number of steps performed with the QL and SARSA methods was higher than the proposal made with the FQL-36 and FQL-20. However, at this point, with the number of training epochs, none of them match the number of steps executed with the deterministic methods. The FQL-36 shows the least amount of deviations towards the loading station, followed by the FQL-20, while the other two methods present a higher number of deviations than this proposal. Table 2 shows the total deviations computed in each scenario.

Likewise, Fig. 13 shows the distribution of the selection of actions made during the simulation. Where the action a3 was the one that was chosen the least times, while the action of going to the destination was the one that was chosen the most times, with these

**Table 2 Number of registered deviations per scenario and method.**

| Method/Scen. | 1 | 2 | 3 | 4 | 5 | 6 | 7 | 8 | 9 | 10 | 11 | 12 | 13 | 14 | 15 |
|---|---|---|---|---|---|---|---|---|---|---|---|---|---|---|---|
| QL | 19 | 76 | 324 | 35 | 154 | 17 | 191 | 84 | 234 | 16 | 9 | 25 | 69 | 8 | 28 |
| SARSA | 187 | 134 | 104 | 12 | 54 | 14 | 110 | 9 | 7 | 59 | 91 | 11 | 27 | 208 | 38 |
| FQL-36 | 7 | 2 | 5 | 2 | 2 | 13 | 7 | 11 | 10 | 10 | 1 | 3 | 3 | 3 | 4 |
| FQL-20 | 6 | 5 | 2 | 3 | 4 | 24 | 22 | 2 | 5 | 4 | 21 | 2 | 1 | 37 | 4 |

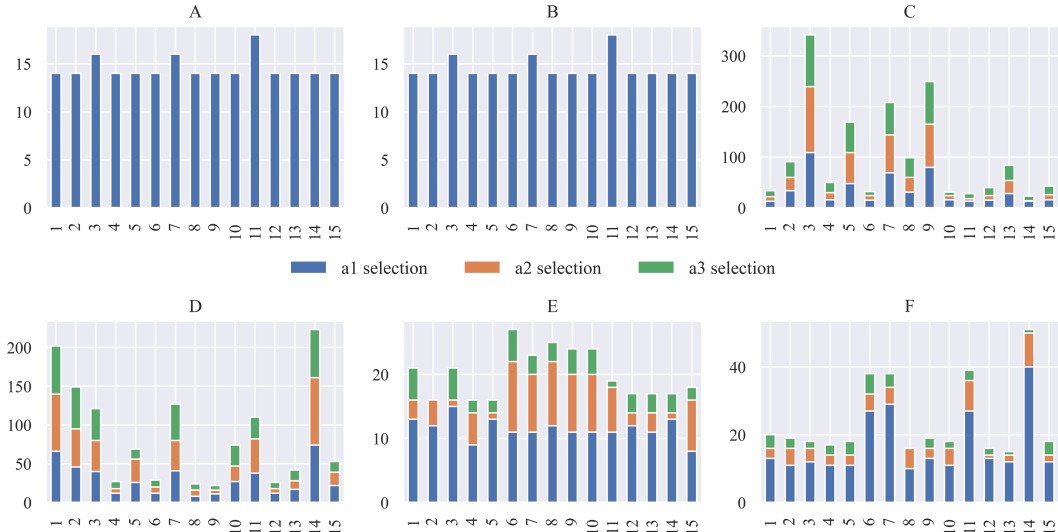

**Figure 13 Distribution of action selection for each scenario concerning the approaches for (A) Threshold, (B) FIS, (C) QL, (D) SARSA, (E) FQL-36 and, (F) FQL-20.**

graphs, it is possible to observe how this selection of actions affects from the point of view that it takes more steps to reach the destination.

Besides, Fig. 14 shows the remaining battery level at the end of the simulations in each of the scenarios. The threshold method and the FIS end up with very similar battery levels. These values correspond to the maximum remaining battery levels attainable in each scenario since these are the methods that perform the fewest steps to reach the destination.

A possible event can occur if the robot changes its direction to the battery charging station and then continues on its way to its destination. This behavior happens with the FQL-20 method in scenarios 6, 7, 11, and 14. Table 2 shows that system FQL-20 deviated from their destination along the 24, 22, 21, and 37 steps. There is a particular advantage in letting the system learn what decisions to make since it makes it more flexible, unlike a deterministic method that could be more rigid.

The spent execution time is a variable that helps to analyze the proposal's performance. Table 3 shows the spent execution time for the simulation to complete the task of going to the goal successfully ten times. The threshold method and the FIS finished with the shortest delay, employing less than 20 ms. While the QL and SARSA methods take the most

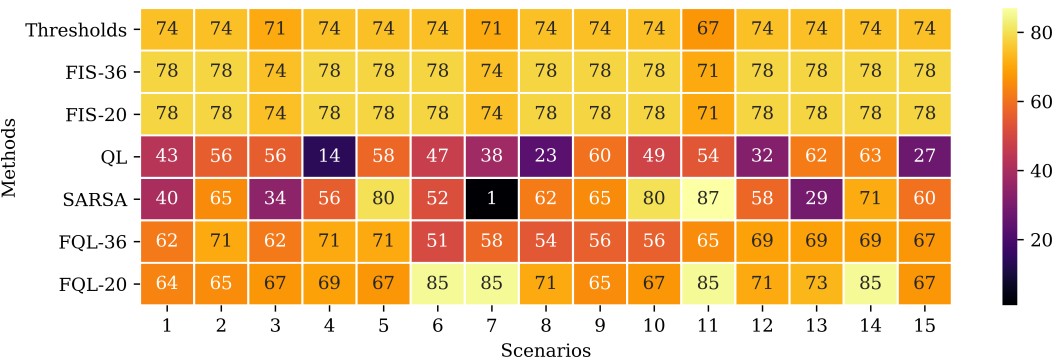

**Figure 14** Remaining battery level at the end of the simulations in each scenario and according to the evaluated approaches.

**Table 3** Simulation execution time to complete the task of going to the goal successfully ten times, for each scenario and according to the evaluated approach.

| | Threshold (ms) | FIS-36 (ms) | FIS-20 (ms) | QL (min) | SARSA (min) | FQL-36 (s) | FQL-20 (s) |
|---|---|---|---|---|---|---|---|
| 1 | 17.60 | 18.42 | 17.96 | 3.53 | 1.93 | 0.36 | 0.26 |
| 2 | 7.91 | 17.87 | 17.65 | 4.85 | 2.91 | 0.33 | 0.35 |
| 3 | 8.00 | 8.52 | 8.48 | 4.42 | 1.26 | 0.26 | 0.26 |
| 4 | 7.63 | 8.29 | 8.39 | 2.83 | 1.29 | 0.26 | 0.26 |
| 5 | 7.73 | 8.36 | 8.21 | 9.16 | 3.99 | 0.26 | 0.23 |
| 6 | 7.75 | 18.16 | 17.75 | 6.75 | 3.24 | 0.83 | 0.51 |
| 7 | 7.92 | 19.42 | 35.61 | 6.01 | 2.29 | 0.80 | 0.40 |
| 8 | 7.87 | 8.32 | 10.44 | 2.97 | 1.64 | 2.15 | 0.28 |
| 9 | 7.76 | 8.38 | 8.36 | 3.72 | 1.76 | 0.44 | 0.29 |
| 10 | 8.14 | 9.61 | 8.28 | 65.38 | 33.63 | 0.90 | 0.30 |
| 11 | 8.15 | 8.78 | 8.81 | 2.61 | 1.15 | 1.19 | 0.43 |
| 12 | 7.80 | 8.26 | 8.30 | 28.91 | 15.64 | 18.70 | 0.27 |
| 13 | 8.46 | 8.32 | 8.41 | 5.94 | 2.36 | 19.76 | 0.26 |
| 14 | 15.31 | 8.26 | 8.37 | 1.56 | 0.98 | 18.47 | 0.34 |
| 15 | 8.44 | 8.12 | 8.64 | 2.40 | 2.04 | 0.60 | 0.30 |

**Table 4** Number of states or rules for each method.

| Method | Threshold | FIS-36 | FIS-20 | QL | SARSA | FQL-36 | FQL-20 |
|---|---|---|---|---|---|---|---|
| Rules/States | 3 | 36 | 20 | 1,030,301 | 1,030,301 | 36 | 20 |

prolonged latency, exceeding 1 min, and in one case, even reaching 65 min. The proposal made in this paper has an intermediate-range execution time of up to 19 s.

One of the reasons why the FQL proposal is advantageous is the reduced number of states that a robot can take during navigation, which is reflected in the memory usage, being this number less than SARSA and QL, as shown in Table 4. This proposal only involves 20 or 36 states, while the QL and SARSA methods, with the same number of entries, involve

**Table 5  Amount of memory occupied by each method.**

| Method | Threshold | FIS approach | QL | SARSA | FQL approach |
|---|---|---|---|---|---|
| Memory (MB) | 48.9 | 49.2 | 121.1 | 121.1 | 49.2 |

1,030,301 states. Table 5 shows the memory usage comparison occupied compared to this article's proposal, obtained with the aid of profiler python, measured throughout the simulated navigation process.

During the development of this work, a validation process was used during the execution of the simulations. The validation procedure consisted of the following: first, after the execution of a new step, it was validated if the coordinates of the simulated robot were at the destination; if so, the task was terminated. Then it was validated if the robot had reached the charging station; if so, the battery charge was restored to 100%, and the simulation continued. Finally, with the robot's coordinates and the obstacles, it was verified if the robot had collided; if so, the simulation indicated a failure and it was restarted; if not, the simulation continued. On average, per scenario, the time taken for validation was 74 ms.

## DISCUSSION

The proposed navigation system enables a mobile robot to move from a starting to a destination point or decide to change its course to go to the battery charging station under a fuzzy rule-based combined with a reinforced learning system. The proposed path planning module enables a robot to move in partial and known environments using a scheme where the robot knows obstacles and senses them, it then plans the path while moving. This module's use provides the system with a reactive behavior in the face of environmental conditions, and thanks to its simplicity, the implementation is painless in embedded systems.

The reactive method introduced in this work allows finding a short path to the destination, although not in all cases since the system can avoid passing through a cluster of obstacles and go more safely. Nonetheless, the method has its limitations. Under certain environmental conditions, the system falls into the local minimum problem, which causes the system to get stuck in a point on the map and not reach destiny. Usually, this occurs when there are groups of obstacles in the form of a fence, and the robot finishes obstructed. Besides, being a global navigation method, the system does not learn, and it must know the destination point, which limits navigation in unknown terrain to a certain degree. In future work, other hybrid methodologies should be explored to avoid falling into the local minimum problem to handle this behavior.

The resultant proposal for decision making based on FQL alongside the path planning module allowed a navigation system with a certain level of autonomy while handling the ARP, given that the proposed system learns which actions to execute, on a trial and error basis, as a function of the input variables, named: the battery level and the distances between the robot and the possible destination points. The tasks chosen within the decision-making module of this proposal were three; the importance of them lies in the fact that the first

action corresponding to the displacement to a destination, which is one of the primary tasks performed by the mobile robot, a human would expect that the robot executes most of the time. Likewise, as the battery charge level was paramount to completing the main task, the second action was to go to a battery charging station. With this, the decision module would seem to be complete. However, a situation could arise in which the robot has a shallow battery level and does not reach the destination or the charging station. This paper considers the third action because a robot should be suspended or shut down to avoid causing damage to its electrical/electronic system. An expert could easily define when to take a particular action. Though, the striking about this work lies in observing the actions selected autonomously by a learning agent. During the simulation, it was possible to observe the decision-making process and how it improved during the training until the agent selected the actions that benefited him complete the journey to the destination. The worst result could be that the robot always remained static; however, the obtained results show that the robot learned that this was not the essential task, and it reached the destination. Nevertheless, it is a behavior that does not occur when observing the results in the proposed scenarios.

During the simulations, the proposed system's behavior does not improve the number of steps required to reach the destination than the threshold method and the FIS, but a significant improvement is observed compared to the QL SARSA methods. This improvement is present in the number of steps executed and the number of training epochs required to achieve ten consecutive successes. During the decision-making stage, the actions that guided the robot to the destination were selected faster than with classic RL methods. Although the classic RL methods resulted in a higher accumulated reward, this did not imply that they solved the task in less time, while the proposal with FQL that had less reward than those methods managed to complete the task in less time. a shorter time. The execution times occupied to complete the route to the destination were more significant than the time taken with the threshold method and the FIS; however, it should note that these methods do not learn, which makes them inflexible compared to the proposal that uses FQL.

It is worth highlighting the result of the FQL-20 proposal, which manages to complete the route to the destination with a higher battery level than with the threshold method and FIS. This result is due to the system's flexibility when it decides to stop at the battery charging station and then continue on its way to its destination. This behavior could be an advantage as long as there are no execution timing restrictions.

On the other hand, the proposal made in this article equates the use of memory to the threshold method and the FIS, which dramatically improves the RL methods. This result shows the system's simplicity in memory usage and its implementation viability in a limited embedded platform like the raspberry pi.

The navigation proposal improved at the action selection phase through simulations, compared with the classical methods of reinforced learning QL and SARSA. The system prioritized and learned to move to a battery charging station when a battery charge reduction was detected over time or went to a predetermined destination. The system's learning depends on the established rules and the reward function that assigns the prizes

and punishments. The reduction in the number of fuzzy rules employed involved an improvement in completing assigned navigation tasks.

For further work, the conducted simulations will be translated to real testing under the considered scenarios throughout the hexapod-type robot's use, proposed in the Introduction Section. Eventually, it will be sought to implement the system in dynamic environments and less structured to take advantage of the six-legged robot's adaptability to verify that the proposal solves the autonomic recharge problem in a mobile robot. It is expected that the computing performance in a limited embedded platform does not degrade drastically.

## CONCLUSIONS

The proposed system allows a mobile robot to have a reactive movement in an environment with dispersed static obstacles and can decide to change the way to the charging station as soon as it considers it necessary or under a critical state to remain static.

System simplification results from using a scheme based on fuzzy rules that allow an expert to limit the number of states in which the robot may fall, dramatically reducing its complexity, compared to the classic reinforced learning scheme that takes each sensed voltage level, or charge percentage point, as a system state. Therefore, consuming more memory space than this proposal. Hence, a rule-based system is advantageous if working with systems with limited computing capabilities.

With this in mind, the following observations arise. The proposed system demonstrates better results than the QL and SARSA methods when observing the number of times it takes to train the system and the number of steps executed with that amount of training epochs. The time that the proposed system takes to complete the task is longer than with the deterministic methods. However, it presents improvements compared to the SARSA and QL methods. It occupies fewer states and requires less memory than the classic RL methods, which is a great advantage considering it will equal the memory used by the threshold-based method and the FIS.

According to the application, the proposal's flexibility, of going to charge the battery and then heading to the destination, can be advantageous as long as time is not the fundamental factor for fulfilling the tasks.

Although this work does not face the energy consumption improvement problem, instead, we try to guarantee the robot's battery availability to complete its task, being this an interesting research topic to develop in future work.

Finally, combining the decision-making module with the FQL method with a path planning module, the decision-making is critical since the generation of a route to the destination usually does not consider the robot's energy levels. In this paper's case, a simple planning method was used that does not have a learning curve and, with the elements of the environment, it generates a path from the resulting forces. It does not consider the battery charge level. However, it would be attractive to explore using a route planning

module that uses deep learning techniques and its implementation in embedded systems in future work.

### Funding

This work was supported by the Instituto Politécnico Nacional (IPN) and Secretaría de Investigación y Posgrado (SIP-IPN) under the projects, SIP20200630, SIP20201397, SIP20200885, SIP20210788, SIP20210124, and SIP20211657, Comisión de Operación y Fomento de Actividades Académicas (COFAA-IPN), also the Consejo Nacional de Ciencia y Tecnología (CONACYT-México) under projects 65 (Frontiers of Science 2015) and 6005 (FORDECYT-PRONACES). The funders had no role in study design, data collection and analysis, decision to publish, or preparation of the manuscript.

### Grant Disclosures

The following grant information was disclosed by the authors:
Instituto Politécnico Nacional (IPN) and Secretaría de Investigación y Posgrado (SIP-IPN): SIP20200630, SIP20201397, SIP20200885, SIP20200569, SIP20210788, SIP20210124, SIP20211657.
Comisión de Operación y Fomento de Actividades Académicas (COFAA-IPN).
Consejo Nacional de Ciencia y Tecnología (CONACYT-México).

### Competing Interests

The authors declare there are no competing interests.

### Author Contributions

- Elizabeth López-Lozada conceived and designed the experiments, performed the experiments, analyzed the data, performed the computation work, prepared figures and/or tables, and approved the final draft.
- Elsa Rubio-Espino conceived and designed the experiments, analyzed the data, authored or reviewed drafts of the paper, and approved the final draft.
- J. Humberto Sossa-Azuela analyzed the data, authored or reviewed drafts of the paper, and approved the final draft.
- Victor H. Ponce-Ponce analyzed the data, prepared figures and/or tables, authored or reviewed drafts of the paper, and approved the final draft.

### Data Availability

The code is available at GitHub: https://github.com/ElizBth/Reactive-Navigation-Under-a-Fuzzy-Rules-Based-Scheme-and-Reinforcement-Learning.git.

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
