# Peer review of "Reactive navigation under a fuzzy rules-based scheme and reinforcement learning for mobile robots"

_PeerJ Computer Science, doi:10.7717/peerj-cs.556_

## Round 0.1 · original submission · Major Revisions

According to the reviewers' comments, this paper has novel information that deserves to be published. However, it is mandatory to address all the reviewers' comments in a new improved version of this paper.

Reviewer 1 ·

Basic reporting

The structure of this paper is not consistent to the objective of the work. In the Introduction section, there should be more focus on the decision-making techniques rather than path-finding.

Experimental design

No comment

Validity of the findings

No comment

Additional comments

Authors published good results. There are many grammatical errors. Focus on the writing on what you want to achieve in the results.

Reviewer 2 ·

Basic reporting

1. This paper has a lot of typing and grammatical errors. The English should be properly reviewed
2. Abstract is very fuzzy, and poor written
3. Paper main contribution is not clear
4. Authors should give more details about implementation, validation procedure, validation time, computational complexity…
5. Authors should justify their improvements against previous results
6. Authors should include technical information in order the reader can replicate the proposed results
7. Authors should include a detailed analysis about energy consumption improvements
8. Authors should include a detailed comparative analysis
9. References should be updated and improved
10. This paper does not present any novelty that deserves to be published

Experimental design

This paper only includes simulation results, it is not clear the importance of selected navigation profiles, and there is no comparative analysis, therefore the proposed research is not rigorous.

Validity of the findings

There is no novelty reported in this paper that deserves to be published, in its actual form this paper appears to be an application of well-known results without any analysis. Authors do not include enough information in order that the proposed results can be replicated. Finally, there is not clear the importance of proposed results for mobile robotics literature.

Additional comments

This paper deals with an interesting topic, however in its actual form research appears to be ambiguous.

Reviewer 3 ·

Basic reporting

No comment.

Experimental design

No comment.

Validity of the findings

No comment.

Additional comments

The paper sounds good, it is well written and structured. Figures have good quality. Simulation results validate their proposal and gives an insight about its performance in contrast with other approaches. Conclusions are coherent and well linked with the results shown. Anyway, this reviewer thinks that it would be interesting if the authors can give an insight about considering another platform instead of the King Spider robot. I mean, what could be the main differences to identify when considering to work with another robot and which criteria must be considered to modify the algorithm in order to adapt it to the new robot platform. In what way the physical/kinematic parameters will impact for the path generation? How this last will affect the path generation? And, if this is the case, how it will affect to the other approaches used? I know that the input variables and fuzzy sets are given in terms of battery level and distances. So, how will impact the use/modification of hardware in your proposal? And with respect to the others approaches? All of these show the same disadvantage/advantage? What about if, instead to go to the charging station, the use of some of the King Spider’s servomotors could be clamped and just those that permit to follow, in some way, the path are limited to use? Maybe another premise variable can be included. This reviewer suggests to include at references section some books about basics on fuzzy logic.

---

## Round 0.2 · accepted · Accept

According to the reviewers' comments, this paper can now be accepted for publication at PeerJ Computer Science

Reviewer 2 ·

Basic reporting

Clear and unambiguous, professional English used throughout. YES

Literature references, sufficient field background/context provided. YES

Professional article structure, figures, tables. Raw data shared. YES

Self-contained with relevant results to hypotheses. YES

Experimental design

This paper only includes simulation results, however methods described with sufficient detail & information to replicate

Validity of the findings

Conclusions are well stated, linked to original research question & limited to supporting results.

Additional comments

All my comments have been addessed, then, I have no more comments

Reviewer 3 ·

Basic reporting

No comment.

Experimental design

This work is limited to simulation results. Anyway, authors present the hardware that can be used for implement their proposal in a future work.

Validity of the findings

No comment.

Additional comments

The authors have addressed all the concerns and comments raised by this reviewer, and most of them have been included in the manuscript. This reviewer has no more comments.